# Sensitivity-Tunable Terahertz Liquid/Gas Biosensor Based on Surface Plasmon Resonance with Dirac Semimetal

**DOI:** 10.3390/s23125520

**Published:** 2023-06-12

**Authors:** Mengjiao Ren, Chengpeng Ji, Xueyan Tang, Haishan Tian, Leyong Jiang, Xiaoyu Dai, Xinghua Wu, Yuanjiang Xiang

**Affiliations:** 1School of Physics and Electronics, Hunan Normal University, Changsha 410081, China; renmengjiao@hunnu.edu.cn (M.R.); jichengpeng@hunnu.edu.cn (C.J.); 202130133005@hunnu.edu.cn (X.T.); haishan_tian@163.com (H.T.); 2School of Physics and Electronics, Hunan University, Changsha 410082, China; xiangyuanjiang@126.com; 3Key Laboratory for Microstructural Functional Materials of Jiangxi Province, College of Science, Jiujiang University, Jiujiang 332005, China; 51205879@163.com

**Keywords:** biosensor, Dirac semimetal, surface plasmon resonance

## Abstract

In this paper, we study the sensitivity-tunable terahertz (THz) liquid/gas biosensor in a coupling prism–three-dimensional Dirac semimetal (3D DSM) multilayer structure. The high sensitivity of the biosensor originates from the sharp reflected peak caused by surface plasmon resonance (SPR) mode. This structure achieves the tunability of sensitivity due to the fact that the reflectance could be modulated by the Fermi energy of 3D DSM. Besides, it is found that the sensitivity curve depends heavily on the structural parameters of 3D DSM. After parameter optimization, we obtained sensitivity over 100°/RIU for liquid biosensor. We believe this simple structure provides a reference idea for realizing high sensitivity and a tunable biosensor device.

## 1. Introduction

An optical biosensor is a kind of micro–nano functional device that can transform a biological signal, which is not easy to measure, into an optical signal, which is easy to observe and measure [1]. Due to the interaction between the light waves and the measured bioanalyte, the slight change in the characteristics of the bioanalyte is represented by the relative obvious change in the parameters of the light signal so as to achieve the purpose of accurately identifying and detecting the characteristics of the bioanalyte or the surrounding environment [2]. Such sensors do not require label or modified biomolecules [3]. As such, they are widely used in biomedicine [4,5], blood detection [6], biochemical detection [7], environmental monitoring [8], food safety [9] and other fields. In recent years, micro–nano scale biosensors have become a research hotspot in the field of biosensors, and many optical biosensor technologies have attracted wide attention, such as fluorescent [10], colorimetric [11], optical fiber [12], evanescent wave photonics [13], etc. In addition, due to the continuous pursuit of high sensitivity and the simple structure of the sensor, many optical sensor structures have been proposed and studied deeply. For example, carbon nanotubes [14], microresonators [15], surface wave imaging [16], photonic crystal band-gap [17] and so on. A biosensor based on surface plasmon resonance (SPR) is a kind of sensor widely studied by researchers, which has certain limitations, such as temperature effect [18] and surface roughness [19]. In addition, the incidence angle must exceed the resonance angle when it is excited. However, compared with traditional optical biosensor schemes, the SPR biosensor has many obvious and unique advantages, such as high sensitivity, real-time detection, easy penetration and no loss [20,21]. Therefore, SPR sensing technology has been widely used in biochemical analysis [22], food safety [23], medical diagnosis [24] and other fields with remarkable effects, such as the detection of staphylococcal enterotoxin in milk [25], drug residue detection [26,27], real-time disease diagnosis [28], gas detection [29], etc. However, the traditional SPR structure usually uses Otto structure or Kretschmann–Raether (KR) structure based on noble metals (such as Au [30], Ag [31], Al [32], etc.) to excite SPR. Although relatively high-sensitivity biosensing can also be realized, the existence of noble metals makes the sensors of these structures maintain shortcomings, such as large inherent loss, limited bandwidth and insufficient dynamic adjustability, which also bring certain limitations to the application of SPR sensors [33,34,35]. In recent years, two-dimensional material graphene has shown great application prospects in the field of sensing due to its good photoelectric characteristics, and it has become a good choice to replace traditional precious metal to excite SPR. Many graphene-based SPR sensor schemes have been proposed [36,37]. It is generally believed that biosensors based on graphene have optimistic prospects in the field of biosensing. However, due to the limitations of the preparation of graphene-based functional devices, the implementation and promotion of graphene-based sensor schemes are also faced with the limitations of process and preparation. Therefore, the exploration of SPR sensing schemes based on new excellent materials and structures is still challenging work.

Recently, in-depth studies of three-dimensional Dirac semimetal (3D DSM) have opened up a new avenue for novel optical biosensors [38]. Three-dimensional DSM is a kind of 3D Dirac material, which has similar electronic and optical properties to graphene and has the possibility of dynamic regulation, while retaining the advantages of metal-like bulk structure, with a certain thickness and relatively simple preparation, and exhibits metal-like properties under certain conditions [39]. In addition, it has beyond the ultra-high charge mobility of graphene, considerable nonlinear plasma performance, longer propagation length and strong light and material interaction. Moreover, 3D DSM is easier to prepare and process than graphene thin films, so it can better make up for the lack of graphene in photoelectric devices. All these provide a new research direction and idea for us to design an SPR biosensor based on 3D DSM. There is no doubt that in terms of surface plasmon resonance, graphene surface plasmon resonance still reigns supreme over most materials discovered to date. However, from the standpoint of the device platform, 3D DSM still has an important role to play. First, for 3D DSM, its bulk structure offers a great degree of flexibility to design the structure parameters. Second, bulk materials are easier to handle than thin films in fabrication facilities, just like how real metals, such as gold and copper, have been successfully used as ingredients in semiconductor foundries. Thus, in nanophotonic applications, we believe that 3D DSM handedly outclasses its metal counterparts and will remain a strong competitor and alternative to graphene-based devices.

Based on this, in this paper, we theoretically propose a terahertz (THz) SPR sensor based on a coupling prism–3D DSM structure. In this structure, a THz SPR sensor with relatively high sensitivity is achieved by combining coupling prism and 3D DSM to excite SPR with 3D DSM instead of conventional noble metals or graphene. In addition, we further found that the conductivity of 3D DSM can be dynamically adjusted by Fermi energy and relaxation time, thus providing a means to control the sensitivity and figure of merit (FOM), and the thickness of the 3D DSM structure has a very significant effect on the sensitivity of the sensor. Through appropriate parameter optimization, we found that when the structure is applied to liquid sensing, the angle sensitivity can reach more than 100°/RIU. When the structure is used for gas sensing, the sensitivity is also relatively high. We believe that SPR tunable optical biosensors based on this structure can find possible applications in biological, chemical, environmental detection and other fields.

## 2. Materials and Methods

### 2.1. Theoretical Model

We considered KR configuration, which consists of coupling prism and the 3D DSM layer, as shown in Figure 1. In this structure, we used polymethylpentene (TPX) as the coupling prism, with a refractive index of $n_{p}=1.46$ [40], and the 3D DSM layer with cadmium arsenide (Cd_3_As_2_), whose refractive index and thickness are denoted as $n_{DSM}$ and $d_{DSM}$, respectively. It is known that the conductivity of 3D DSM is electrically tunable, so by adding 3D DSM between the coupling prism and the sensing layer, the sensing performance of the whole structure can be dynamically tunable. The Boltzmann transport equation (BTE) is a partial differential equation that describes the statistical behavior of a thermodynamic system in a state of non-thermodynamic equilibrium. For the 3D DSM structure, under the relaxation time approximation, there is a semiclassical Boltzmann transport equation, namely $\tau (\varepsilon_{k})=\tau$; therefore, the linear electrical conductivity and refractive index of 3D DSM can be expressed as [41]:(1)σDSM(ω)=σ043π2τ1−iωτ(kBT)2ℏ2vF2Li2(−e−EFkBT)+(EFkBT)2+π23,
(2)$$n+ik=\sqrt{1+i\sigma_{DSM}/\varepsilon_{0}\omega },$$
where ω is the angular frequency of the incident light; $E_{F}$ is the Fermi energy; τ represents the relaxation time; ℏ is the reduced Planck constant; $k_{B}$ and T are the Boltzmann constant and temperature, respectively; $v_{F}$ is the Fermi velocity; $Li_{s}(z)$ is the polylogarithm; and $\sigma_{0}=e^{2}/4\hbar$. In the next calculation, the original parameter of the 3D DSM structure is set as $E_{F}=0.1\;\mathrm{eV}$ and τ=1.0 ps, and the incident light frequency is 1THz.

### 2.2. Methods

In order to obtain the reflectance of the whole structure, the relatively mature transfer matrix method is adopted in this scheme. When a light beam is incident on a metal surface, the plane where the normal line between the incident light and the metal surface lies is the incident surface. When the electric vector of the incident light is perpendicular to the plane of incidence, the polarized light is called S-polarized light or TE wave. Polarized light whose electric vector is parallel to the plane of incidence is called P-polarized light or TM wave. We know that SPR can only be excited under TM polarization. Therefore, we only need to consider the case of TM polarization. Firstly, the transfer matrices at the junction between the coupling prism and 3D DSM and at the junction between 3D DSM and the upper sensing layer in the structure shown in Figure 1 are, respectively:(3)$$D_{p\to d}=\frac{1}{2}\left[\begin{array}{cc} 1+\eta_{1}+\xi_{1} & 1-\eta_{1}-\xi_{1}\\ 1-\eta_{1}+\xi_{1} & 1+\eta_{1}-\xi_{1} \end{array}\right],$$
(4)$$D_{d\to s}=\frac{1}{2}\left[\begin{array}{cc} 1+\eta_{2}+\xi_{2} & 1-\eta_{2}-\xi_{2}\\ 1-\eta_{2}+\xi_{2} & 1+\eta_{2}-\xi_{2} \end{array}\right],$$
where $\eta_{1}=\varepsilon_{p}k_{dz}/\varepsilon_{d}k_{pz}$, $\xi_{1}=\sigma k_{dz}/\varepsilon_{0}\varepsilon_{d}$, $\eta_{2}=\varepsilon_{d}k_{sz}/\varepsilon_{s}k_{dz}$, and $\xi_{2}=\sigma k_{sz}/\varepsilon_{0}\varepsilon_{s}$. Additionally, $k_{iz}$ denotes the component of the wave vector $k_{i}$ in the z direction, $k_{i}=\sqrt{\varepsilon_{i}}\omega /c$. c is the speed of light in vacuum, $\varepsilon_{0}$ is the vacuum dielectric constant and θ is the incident angle. When an electromagnetic wave propagates in a uniform 3D DSM medium with a thickness of $d_{DSM}$, it can be expressed as the following propagation matrix:(5)$$P_{DSM}(d_{DSM})=\left[\begin{array}{cc} e^{-ik_{z}d_{DSM}} & 0\\ 0 & e^{-ik_{z}d_{DSM}} \end{array}\right].$$

For the structure shown in Figure 1, the transfer matrix of the whole structure can be expressed as:(6)$$M=D_{p\to d}P_{DSM}D_{d\to s},$$

As a result, the reflected coefficient is as follows: $r=M_{21}/M_{11}$. Based on the above, we can finally obtain the reflectance of the structure as follows:(7)$$R_{p}={\left|r_{}\right|}^{2}.$$

For a sensor, the core indicator to measure its performance includes sensitivity, half-wave full width (FWHM), the FOM and so on. Because this paper mainly studies the slight shift in the angle corresponding to the SPR reflected peak caused by the slight variation in the refractive index of the sensing layer, the angle sensitivity is used to measure the sensor, which is specifically expressed as:(8)$$S_{\theta }=\frac{\Delta \theta }{\Delta n_{}},$$
where Δθ is the offset of the formant angle and represents the change in the refractive index. It is worth mentioning that sensitivity can also be represented by intensity sensitivity. Considering continuity with our previous work, we still use the representation of angle sensitivity here. In addition, for the FOM, its calculation expression is: $FOM=S_{\theta }\cdot DA$, where DA=1/FWHM.

## 3. Results and Discussion

### 3.1. The Effect of 3D DSM

In this section, we discuss the sensing characteristics of the SPR THz biosensor based on the structure shown in Figure 1. As we know, it is a standard method for SPR biosensors to perceive small changes in the characteristics of sensing media (such as refractive index, etc.) by observing the changes in the reflected peaks. In this paper, we assume the electromagnetic wave incident from the prism at the angle of θ and plot the change in reflectance with the angle of incidence when there is 3D DSM or not, as shown in Figure 2a. It can be found from Figure 2a that when there is no 3D DSM, total internal reflection occurs in the part where θ is greater than 66°, while after the addition of 3D DSM, a relatively obvious reflected peak can be observed near θ=66.5°, which is caused by SPR excitation. The confirmation of SPR can be obtained by the following dispersion relation [41]:(9)$$\coth\left(\sqrt{q^{2}-\varepsilon_{DSM}\omega^{2}/c^{2}}\frac{d_{DSM}}{2}\right)=-\frac{\varepsilon_{DSM}}{\varepsilon_{d}}\frac{\sqrt{q^{2}-\varepsilon_{d}\omega^{2}/c^{2}}}{\sqrt{q^{2}-\varepsilon_{DSM}\omega^{2}/c^{2}}},$$
where $d_{DSM}$ represents the thickness of 3D DSM. Because the dispersion curve is reflected in other literature [41], it is not described here. When the structure is used for liquid sensing, we assume that the semi-infinite background material above the 3D DSM structure corresponds to an aqueous solution with a refractive index of $n_{s}=1.33$. Such an aqueous solution can be realized in practical experiments by constructing a flow pool above the 3D DSM layer. Cd_3_As_2_ is adopted for the 3D DSM layer, corresponding to the original thickness, Fermi energy and relaxation time, i.e., $d_{DSM}=1300\;\mathrm{nm}$, $E_{F}=0.1\;\mathrm{eV}$ and τ=1.3 ps, respectively. At this time, it can be clearly observed that a sharp reflected peak appears in the reflectance curve near 66.1°, which is a typical feature of SPR excitation. In order to more intuitively observe the impact of small changes in environmental parameters on the reflectance of the entire structure, people often define sensitivity as the change rate of the angle of the reflected peak (the minimum value of reflectance) relative to environmental parameters (such as refractive index). This is also a common practice in this field.

When the refractive index of the sensing medium changes slightly due to the change in the environment, the reflected peak changes accordingly. We adopt the widely used refractive index variation Δn=0.005, that is, when the refractive index changes to $n_{s}=1.335$ [42,43,44,45], the reflected peak moves from the initial 66.61° to 67.12°, and a minor change in the refractive index causes a slight shift in the reflected peak. According to the Expression (8) mentioned in the previous section, we can calculate that sensor sensitivity can reach 102 °/RIU at this time. It can be seen that the reflected peak in the reflectance curve of the structure is very sensitive to small changes in the refractive index of the sensing layer, and the SPR sensor scheme based on the structure is feasible.

### 3.2. The Effect of Refractive Index on Formant Angle

We know that when the incidence angle is greater than the resonance angle, SPR is excited on the surface. Therefore, for different refractive indices of the sensing layer, the formant angle also changes accordingly. From that, we plotted the curve of the formant angle changing with the refractive index, as shown in Figure 2b. According to the Expression (8) mentioned in Section 2, the slope in this figure represents the angular sensitivity of the sensor. As can be seen from the figure, the formant angle shows a linear upward trend with the change in the refractive index. The line becomes a little steeper as it grows, and its slope increases slightly. The sensitivity of the sensor changed from an initial 99.6°/RIU to approximately 101.6°/RIU. It can be seen that the sensitivity of the sensor to the change in refractive index is obvious, and the sensitivity gradually increases in the range of 1.33~1.335. The additional data points between 1.330 and 1.335 examined that the sensor responds linearly. The calibration curve method is a typical method for determining the sensitivity performance of the refractive index [46,47].

### 3.3. The Effect of Different Thickness of 3D DSM

Combined with the previous theory, we know that the reflected peak of SPR is related not only to the structural parameters of the material, but also to the characteristics of the material itself. Furthermore, the influence of the change in these parameters on the sensitivity plays a very critical role, which provides a key basis for us to design a reference scheme with higher sensitivity. A such, we focus on the influence of each material and structural parameter on the sensitivity of the biosensor in the following discussion. In addition, in order to more intuitively observe the impact of small changes in the environmental parameters on the reflectance of the entire structure, people often define sensitivity as the change rate of the angle of the reflected peak (the minimum value of reflectance) relative to environmental parameters (such as refractive index). This is also a common practice in this field. First, we consider the reflectance curve and its sensing performance under different thicknesses of 3D DSM. We plotted the curve of the sensor reflectance as a function of the incident angle when the thickness of 3D DSM is $d_{DSM}=1300\;\mathrm{nm}$, $d_{DSM}=1700\;\mathrm{nm}$, $d_{DSM}=2100\;\mathrm{nm}$ and $d_{DSM}=3000\;\mathrm{nm}$, as shown in Figure 3. It can be found that the reflectance curves corresponding to the thickness of 3D DSM varying within this range all have corresponding reflected peaks, that is, the excitation of SPR is not be affected, but the sharpness is slightly different, which affects the FWHM of the reflectance curve, and then affects the FOM of the sensor. Therefore, on this basis, we plotted the changing trend curve of various related sensor performance parameters corresponding to the gradual change in the 3D DSM structure’s thickness from $d_{DSM}=1300\;\mathrm{nm}$ to $d_{DSM}=3000\;\mathrm{nm}$, as shown in Figure 4. It can be clearly seen from the curve results that when the thickness of the 3D DSM structure is within the range of 1300~3000 nm, the FOM rises and the sensitivity decreases with the increase in thickness, which has a good guiding role for our subsequent parameter selection. According to the above analysis, lower thickness of 3D DSM make the sensor exhibit higher sensitivity. Therefore, we try to take a thinner 3D DSM, and to facilitate the calculation, we fixed its thickness to be $d_{DSM}=1300\;\mathrm{nm}$ in the subsequent design. On this basis, we seek other regulation laws.

### 3.4. The Effect of Different Fermi Energy of 3D DSM

Next, we further discuss the influence of material parameter properties of 3D DSM on the sensing properties of the whole structure. Material parameter properties can not only help us to find relatively high-sensitivity sensing schemes, but also provide a very key reference for achieving dynamic and tunable sensitive characteristics. We first pay attention to the impact of the change in 3D DSM Fermi energy on the sensing performance of the whole structure, as shown in Figure 5. When the Fermi energy is $E_{F}=0.09\;\mathrm{eV}$, $E_{F}=0.13\;\mathrm{eV}$, $E_{F}=0.16\;\mathrm{eV}$ and $E_{F}=0.19\;\mathrm{eV}$, the curves of reflectance changing with the incident angle are obtained, and the corresponding sensor sensitivity at this Fermi energy is also calculated. As can be seen from Figure 5, its sensitivity decreases with the increase in Fermi energy in the range of 0.09~0.19 eV. However, by observing the shape of its reflected peak, the reflected peak is sharper when $E_{F}=0.13\;\mathrm{eV}$, so we adjust the Fermi energy to $E_{F}=0.13\;\mathrm{eV}$. In addition, we found in the calculation that the trend curve of the sensor performance changing with Fermi energy is similar to that of different 3D DSM structure’s thicknesses (Figure 4), and the relevant change law can be referred to in Figure 4.

### 3.5. The Situation for Gas

In the above discussion, we selected the parameters of the sensor for liquid. Through parameter optimization, we can further explore the sensing performance when the sensing medium is gas. We set the original parameters as $d_{DSM}=1300\;\mathrm{nm}$ and τ=1.3 ps, and considered the sensitivity of the gas sensor when the refractive index of the sensing medium changed from $n_{s}=1.00$ to $n_{s}=1.005$ [48]. Here, we only considered the influence of Fermi energy and plotted a comparison of reflectance curves at different Fermi energy rates, as shown in Figure 6. After comparison, we found that the formant angle of the gas sensor decreased compared to that of the liquid sensor. At the same time, with the increase in Fermi energy, the reflected peak became sharper and sharper, while the corresponding sensor sensitivity showed a trend in gradual decrease. Moreover, after certain parameters’ optimization, we obtained a gas sensor with better performance.

### 3.6. Other Notes to This Article

It is worth mentioning that the experimental results and data related to the work in this paper are only in the theoretical research stage and need to be confirmed by practical applications. Indeed, compared with theoretical work, experimental verification involves more practical issues. Although current processing technology, preparation technology and measurement methods can also support the experimental verification in the manuscript, considering the relatively complex requirements for experimental conditions, in this work, we only focus on the improvement in SPR sensor sensitivity and other performance parameters from a theoretical point of view.

Furthermore, our work is theoretical based on numerical calculations, so we focus on the theoretical performance parameters of sensitivity-tunable sensors based on the 3D Dirac semimetal (3D DSM) structure. Three-dimensional DSM is the general name given to a class of materials. In order to facilitate calculation, Cd_3_As_2_, a typical 3D DSM material, is adopted in this paper. In order to solve the safety problems of Cd_3_As_2_, we propose two solutions in the actual experiment: one is to take corresponding isolation measures in the actual experiment; the other is to adopt 3D DSM materials that are similar to Cd_3_As_2_ but not toxic based on the principle that different refractive index values correspond to different materials. Therefore, in our follow-up work, the safety of the materials used will be our further focus.

Our research group has carried out similar work before, which is to study the highly sensitive sensor based on the 1D PC and the graphene-structure-excited optical Tamm state (OTS) [49], and the sensitivity of this work is indeed higher compared to this work. However, there are completely different mechanisms and structures between the two. From a sensitivity perspective alone, Tang’s work has higher sensitivity, but the composite structure is complex. Moreover, the preparation and transfer of graphene are difficult, and the fabrication of micro–nano structures is complex. However, the structure proposed in this manuscript is simple, and the processing technology and material preparation are easier. Therefore, our solution has a higher competitive advantage and wider practical applications under conditions that require lower costs and do not have strict sensitivity requirements.

Indeed, the refractive index range of 1.33–1.335 is mainly targeted at the refractive index of the solution. For solutions with different concentrations, their refractive index varies depending on the concentration. The refractive index of solutions with different concentrations generally increases gradually from 1.33. Some of the literature shows the variation in solution refractive index with concentration in some schemes [50]. In addition, the refractive index of many solutions (such as heavy metal ions solutions [51] and glucose solutions) generally ranges from 1.33 to 1.335. As a theoretical model calculation, we set refractive indices of 1.33 and 1.335 in this work. This is also a common practice in many references [42].

## 4. Conclusions

In summary, we propose a THz sensor scheme based on SPR. This scheme excites SPR by a coupling prism and 3D DSM, generating sharp reflected peaks. Meanwhile the introduction of 3D DSM significantly improves the sensor sensitivity, and it also provides a means to dynamically adjust its sensing characteristics. The calculation results show that the sensing performance of the SPR biosensor is not only related to the structural parameters, but also closely related to the material parameters of 3D DSM. When the sensor is applied to liquid sensing, through structure and parameter optimization, we can obtain sensitivity of over 100 °/RIU. Using a commercial encoder angle resolution of 0.01 degree, the proposed 102°/RIU has a theoretical refractive index resolution of 9.8 × 10^−5^. At the same time, when the structure is used for gas sensing, good sensing performance is also obtained. Compared with the traditional SPR biosensor, this scheme has a simpler structure, lower production process requirements and relatively high sensitivity. We believe this scheme is expected to show potential applications in the field of micro–nano structure-based biosensing.

## Figures and Tables

**Figure 1 sensors-23-05520-f001:**
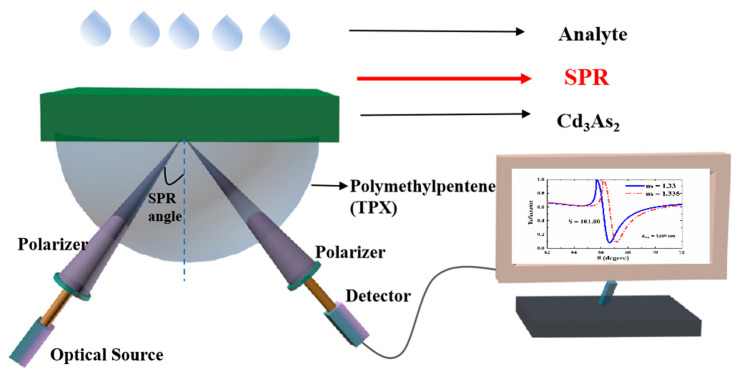
Schematic diagram of SPR sensor based on coupling prism and 3D DSM.

**Figure 2 sensors-23-05520-f002:**
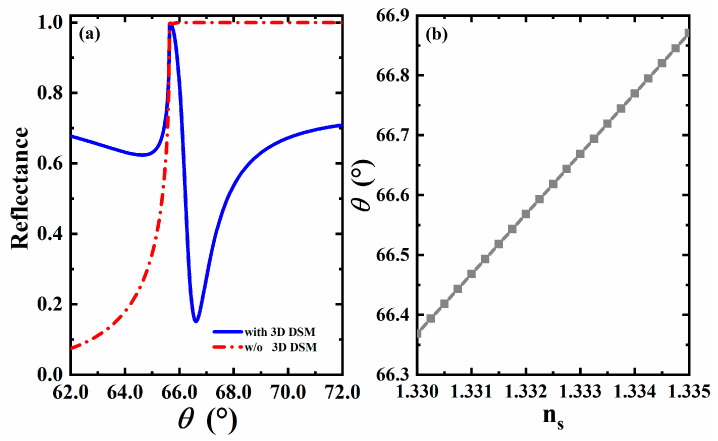
(**a**) Comparison plot of reflectance with incident angle with (blue line) and without 3D DSM (red line); (**b**) Curve plot of formant angle changing with refractive index.

**Figure 3 sensors-23-05520-f003:**
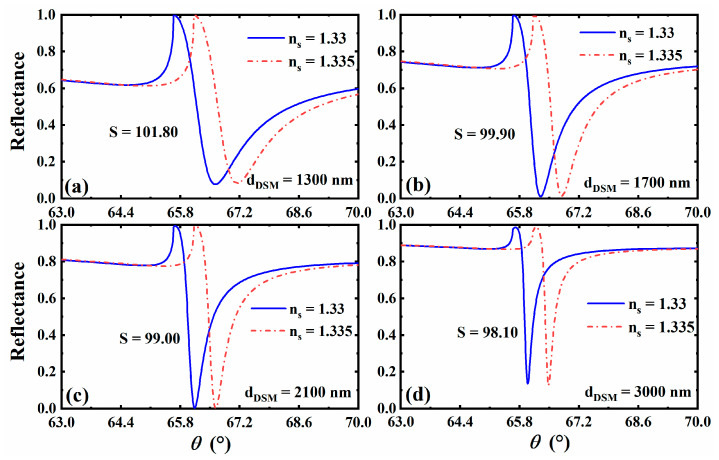
The reflectance curve with the incident angle when the thickness of 3D DSM is taken as $d_{DSM}=1300\;\mathrm{nm}$ (**a**), $d_{DSM}=1700\;\mathrm{nm}$ (**b**), $d_{DSM}=2100\;\mathrm{nm}$ (**c**), and $d_{DSM}=3000\;\mathrm{nm}$ (**d**).

**Figure 4 sensors-23-05520-f004:**
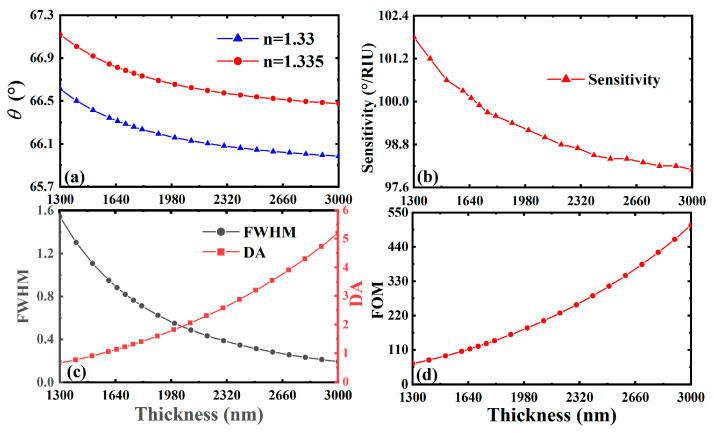
The formant angle (**a**), sensor sensitivity (**b**), FWHM and DA (**c**), and FOM (**d**) with different thicknesses of 3D DSM.

**Figure 5 sensors-23-05520-f005:**
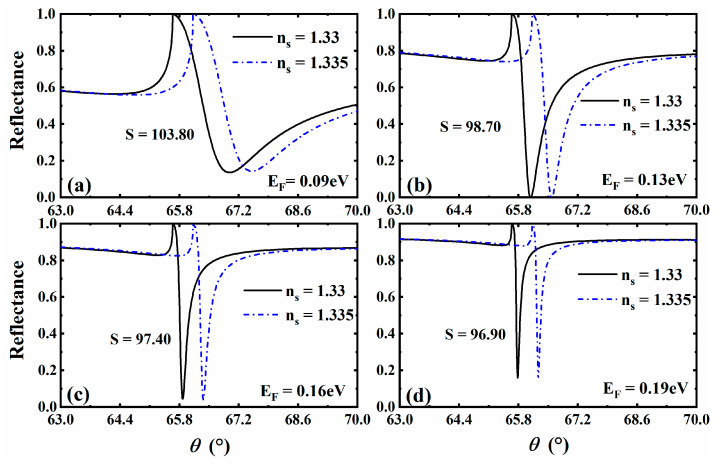
Curve of reflectance as a function of incident angle when the Fermi energy of 3D DSM is taken as $E_{F}=0.09\;\mathrm{eV}$ (**a**), $E_{F}=0.13\;\mathrm{eV}$ (**b**), $E_{F}=0.16\;\mathrm{eV}$ (**c**), and $E_{F}=0.19\;\mathrm{eV}$ (**d**).

**Figure 6 sensors-23-05520-f006:**
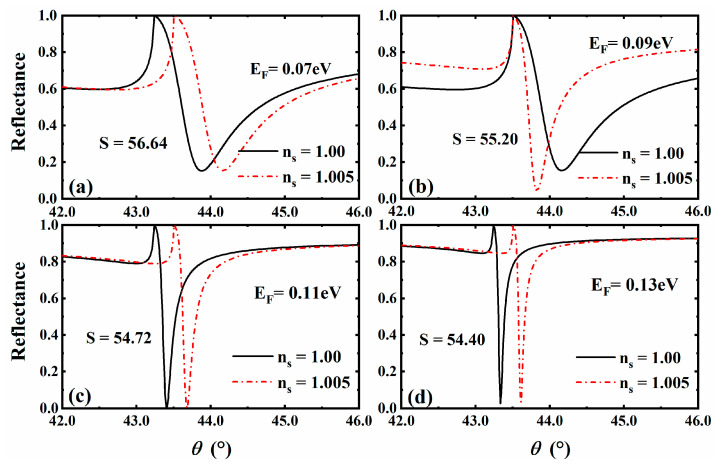
Curve of the reflectance with the angle of incidence at Fermi energy of 3D DSM: $E_{F}=0.07\;\mathrm{eV}$ (**a**), $E_{F}=0.09\;\mathrm{eV}$ (**b**), $E_{F}=0.11\;\mathrm{eV}$ (**c**), $E_{F}=0.13\;\mathrm{eV}$ (**d**).

## Data Availability

The data presented in this study are available on request from the corresponding author.

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
