# Peer review of "Sensitivity-Tunable Terahertz Liquid/Gas Biosensor Based on Surface Plasmon Resonance with Dirac Semimetal"

_sensors, 2023, doi:10.3390/s23125520_

Round 1
Reviewer 1 Report
The authors presented theoretical analysis using transfer matrix method of one layer sandwiched between a polymer substrate and a cladding that varies. The sensing layer complex refractive index was calculated from an analytical model using Boltzmann transport equations. The optical properties (refractive index and electrical conductivity) are calculated when selecting Fermi energy level and response time. The authors used this model to study the effect of varying the film thickness and Fermi energy on the angular spectrum response of the proposed SPR device. The work has merits in it. However, there is some good room for improvement to bring the paper to the final acceptable form. Please find below my comments:
1- In section 2, theoretical model. The authors mentioned optical conductivity, that should be electrical conductivity. In general when dealing with complex index, the imaginary part can be either placed in conductivity or in the attenuation coefficient.
2- In the same section. It would be great if the authors added few lines to explain the main concept of usinnng Boltzmann transport equation in order to obtain equations 1 and 2.
3- The line after equation 6 is missing the definition of reflectance coefficient r.
4- In the results and discussion the authors focus on the peak in the angular spectrum. This is however is misconception. SPR occurs when the photon energy is transferred into charge oscillation at the surface.Hence, at resonance the photon will be absorbed and a dip is expected in the reflectance as clearly seen in the figures. Hence, the analysis in this section should mainly focus on the dip location. True the peak is an anomaly however from the basic definition of SPR the dip is what reflects conceptually this effect.
5- The authors studied the effect of the film thickness on the SPR signal. The narrowing of the observed spectrum can be attributed to the increase in absorption. This is also the case for increasing the Fermi energy. However, the change in the dip angle remains to be explained. I appreciate if the authors could shed some light on this effect.
There are several places in the text where the authors use long or overrun sentences. A minor revision is needed.
Reviewer 2 Report
This is a review report for the manuscript titled "Sensitivity-Tunable Terahertz Liquid/Gas Biosensor Based on Surface Plasmon Resonance with Dirac Semimetal" by Mengjiao Ren, et al. submitted to Sensors.
The manuscript proposed a theoretical terahertz surface plasmon resonance (SPR) sensor device for water or atmospheric gaseous sensing. A three-dimensional Dirac semimetal (3D DSM) structure consisting of a coupling prism is proposed and analyzed. Based on the authors' numerical data, the theoretical SPR sensor achieved good responsiveness (S_max = 101.80 degree/RIU) to a step change in the water-like refractive index between 1.330 and 1.335. On the other hand, S_max drops significantly to 56.64 degree/RIU when the step change varies between 1.000 and 1.005.
Comments for the authors:
1) The proposed work mainly focuses on examining the Fermi energy theoretically of the SPR sensor with a different material. It will be clearer to write the intention in the introduction sections as there are other critical parameters to look out for when assessing the SPR sensor. For example, losses due to material(s) used to excite the surface plasmons.
2) In the Introduction section, the advantages of the SPR sensors are described covering different techniques and applications. It is advisable to relate to the readers that there are limitations of SPR sensors, such as temperature effect, and surface roughness that are not extensively covered in the manuscript. A discussion on limitations will be very helpful in the manuscript by providing references to recent review articles, in new paragraphs or tables.
3) There seems to be missing information (incomplete sentence) in the manuscript. The authors are kindly reminded to proofread before submitting the manuscript. i.e.:
4) Line 80: "Through appropriate parameter optimization, we found that when the structure is applied to liquid sensing, the angle sensitivity can reach more than."
5) Line 126: "as a result, the reflected coefficient is: ."
6) In the Results and Discussions section, the sensitivities to a change in refractive index for Figure 3 and 4 are calculated using equation (8), on line 134. It will greatly provide more insights to readers if the authors could provide another graph with more data points in between the two refractive indice. i.e. like 1.331, 1.332, 1.333 and so on. This new graph (x axis refractive indice, y axis degree) could show the linearity parameters, for example, to support the high sensitivity conclusion.
7) The sensitivity to the refractive index in the manuscript is described in degree per RIU. It will be easier for direct comparison if the unit is also provided in RIU resolution. The approximated RIU resolution using the typical degree resolution available in the market will be helpful to the work.
The authors could advise more information referring to the following paper found online:
8) This submitted manuscript with slight variations in words minus the formatting can also be found online (https://www.researchgate.net/publication/369974810_Sensitivity-Tunable_Terahertz_LiquidGas_Biosensor_Based_on_Surface_Plasmon_Resonance_with_Dirac_Semimetal) as 'preprint' in the public domain.
9) Referring to another recently published paper by the authors with similar work using graphene-based SPR sensor: High-Sensitivity Terahertz Refractive Index Sensor in a Multilayered Structure with Graphene (Published: 10 March 2020), Nanomaterials 2020, 10, 500. DOI: 10.3390/nano10030500
The current manuscript has lower sensitivity when operating in the terahertz. Between the manuscript and this published work, which has more promising for practical application? The author should include the comments and add as the references in the introduction or conclusion for discussions.
Other minor comments:
10) Figure 2: The red dashed-dotted line at y=1.0 is not noticeable-visible. Extending the y-axis upper limit to for example 1.1 in the plot will be clearer. And have a pointing arrow indicating the disappearance of any total internal reflection at 66 degrees.
I have my comments that will be useful for you.
Reviewer 3 Report
The authors propose a theoretical study of a polymer prism with a toxic CdAs layer and measurements in the THz range. The work is interesting, but there remain some questions as noted below to make the paper more practically useful for experimentalists and allow for more citations. Furthermore, there are some errors that should be corrected as noted below.
1. Introduction: sentence 1 and 2 have no references.
2. “In addition, due to the continuous pursuit of high sensitivity and simple structure of the sensor, many optical sensor structures have been proposed and studied deeply.” This sentence should have three references.
3. Page 2 line 48-52 misses references
4. Page 2 line 81: unfinished sentence
5. TM not defined. In general, all abbreviations should be better defined before the first use throughout the whole manuscript.
6. Page 3 line 126: coefficient is missing.
7. Based on the methods section the work seems to be fully theoretical, since no devices, methods to produce THz frequency and sensors and production methods are mentioned. The outline that the work is theoretical is missing.
8. Why do the authors propose highly toxic substances as CdAs? In biological applications this is regarded unsuitable for e.g. implants but also a problem during production (work safety). How do the authors propose a real production? I know a student that got poisoned from Cd.
9. In accordance to the comment before, which ISO standards are dealing with Cd and a safe work that therefore could allow a safer production? This should be added and discussed as well.
10. Page 4 line 150: space sign missing – “partwhere“.
11. Page 4 line 156: other literature is not referenced here and should be cited.
12. Figures 2, 3, 4, 5 and 6: The angle theta should be in italic as well in the diagram. Moreover, the degree sign should be used (Alt + 0176).
13. Free lines between a figure and manuscript text is missing.
14. Although the work proposes some sensitivity, to use it practically, non-physicists usually do not calculate amounts in fermi level. The authors should provide a sensitivity in more useable parameters like layer thickness, or concentrations as well.
15. The authors should relate their results with works that are common for surface analysis like QCM[1] or IR surface reflection techniques[2].
16. The authors contribution should use the taxonomy of the publisher: https://www.mdpi.com/data/contributor-role-instruction.pdf
References:
[1] M. Schönhoff, P. Bieker, Macromolecules 2010, 43, 5052. DOI: 10.1021/ma1007489
[2] A. Frueh, S. Rutkowski, I. O. Akimchenko, S. I. Tverdokhlebov, J. Frueh, Appl. Surf. Sci. 2022, 594, 153476. DOI: 10.1016/j.apsusc.2022.153476
Mostly missing space signs and free lines are a problem here.
Round 2
Reviewer 2 Report
This is the second review report for the manuscript titled "Sensitivity-Tunable Terahertz Liquid/Gas Biosensor Based on Surface Plasmon Resonance with Dirac Semimetal" by Mengjiao Ren, et al. submitted to Sensors.
The authors have revised the manuscript and attached a cover letter describing the changes made to the original manuscript. The authors have made replies to all the comments. However, there are additional comments to be addressed:
In response to question 6 in the first review report, the authors have replied and added a subplot in Figure 2 to show the change in angle with changes to the refractive index.
1) The graph does not show any data point's symbol indicating the points of the plot.
2) The purpose of additional data points between 1.330 and 1.335 are to examine whether the sensor will respond linearly or not. This method of calibration curves is to determine the refractive index sensitivity performance, citing the similar method from established biosensor-SPR related literature. Example:
- "Sensitivity Analysis of Single- and Bimetallic Surface Plasmon Resonance Biosensors", Sensors. DOI: 10.3390/s21134348
- "A plasmonic refractive index sensor with an ultrabroad dynamic sensing range", scientific reports. DOI: 10.1038/s41598-019-41353-4
In response to question 7 in the first review report, the authors have replied with their explanations.
3) When comparing performance between different literature the use of the RIU resolution is very common in high impact/established publications. For examples,
- "Enhanced Sensitivity of Anti-Symmetrically Structured Surface Plasmon Resonance Sensors with Zinc Oxide Intermediate Layers", Sensors. DOI: 10.3390/s140100170
- "A plasmonic refractive index sensor with an ultrabroad dynamic sensing range", scientific reports. DOI: 10.1038/s41598-019-41353-4
- "Advances and applications of nanophotonic biosensors", Nature Nanotechnology. DOI: 10.1038/s41565-021-01045-5
The authors might have misunderstood that all the SI unit has to be changed from the first review report. These are not the suggested changes. If the authors have read again on other research group literature on refractive index sensors, they would notice that the literature has indicated in the conclusion or abstract their sensor's RIU resolution. For example, using a commercial xxx brand encoder angle resolution of 0.01 degree, the proposed 102 degree/RIU will have a theoretical refractive index resolution of 9.8x10^-5. I believe having this information will be very beneficial to the manuscript and research community.
4) In the manuscript and replies, the authors have emphasised on the use case of biosensor. Have the authors taken any practical approach to explain why 1.330 and 1.335 refractive index were chosen? Any cited references? Bioanalyte has a wide range of refractive index, ~1.330 seem limited to water based biomed use cases.
5) In line 198 "we can calculate that the sensor sensitivity can reach 102 degree/RIU at this time". I would assume the authors have made the following calculation (67.12-66.61)/0.005=102 degree/RIU. Given that the data for questions 1) and 2) are not available, I would not support this calculated refractive index sensitivity at this point.
I hope the authors would take my comments in a good way to improve the manuscript.
Reviewer 3 Report
The authors of the manuscript with the title: Sensitivity-Tunable Terahertz Liquid/Gas Biosensor Based on Surface Plasmon Resonance with Dirac Semimetal improved the manuscript according to the reviewer comments. I am satisfied with the answers given by the authors. Therefore, I would like to suggest the editor to accept the paper for publication in the journal Sensors.
